# A Comparison of the Impact of Two Methods of Nutrition-Behavioral Intervention on Selected Auxological and Biochemical Parameters in Obese Prepubertal Children—Crossover Preliminary Study

**DOI:** 10.3390/ijerph16162841

**Published:** 2019-08-08

**Authors:** Agnieszka Kozioł-Kozakowska, Małgorzata Wójcik, Aleksandra Furtak, Dominika Januś, Jerzy B. Starzyk

**Affiliations:** 1Department of Pediatrics, Gastroenterology and Nutrition, Institute of Pediatrics, Jagiellonian University Medical College, 30-663 Krakow, Poland; 2Children’s University Hospital in Krakow, 30-663 Krakow, Poland; 3Department of Pediatric and Adolescent Endocrinology, Chair of Pediatrics, Institute of Pediatrics, Jagiellonian University, Medical College, 30-663 Krakow, Poland

**Keywords:** weight reduction, children and adolescents, obesity, physical activity

## Abstract

Obese children are exposed to short and long-term health consequences, such as dyslipidemia, hypertension and diabetes mellitus. For these reasons, the prevention and treatment of obesity in the pediatric population is a challenge for health care professionals. The aim of this study was to evaluate whether an intensive intervention based on diet and physical activity has a better impact on the auxological and biochemical parameters than standard care (intervention). The study included 20 children (six boys, 14 girls), of the mean age 8.9 (SD 1.4) before puberty. The participants were randomly assigned to two groups: Group I (starting treatment with intensive intervention), and II (starting treatment with standard intervention). After three months, the groups were switched. The comparison of the two interventions in the study group indicates a better effectiveness of intensive intervention in the improvement of anthropometric parameters and majority of biochemical ones (except for insulin concentration, HOMA IR index and LDL cholesterol). As the result of intensive intervention, the mean % of weight-to-height excess and hip circumference decreased significantly (*p* < 0.005). Our results confirm that complex intervention based on systematic control visits, including personalized dietitian counselling and physician care, during the weight reduction process is more effective than a one-off standard visit.

## 1. Introduction

Obesity is a global problem. The increase in the number of obese adults is a consequence of increasing prevalence of obesity among children [1,2]. The incidence of childhood obesity has increased two-to-three times in the last two decades. Currently in the world, approximately one in six or 12.7 million children and adolescents between 2 and 19 years of age are overweight or obese [3]. In the Polish population of children aged 13–36 months, overweight concerned 14.5% and obesity could be designated to 13% of children and, among preschoolers, respectively, 10.4% and 2.3% [4,5]. Excess weight among eight-year-olds included in the Childhood Obesity Surveillance Initiative (COSI) was found in almost 1/3 of children (30.7%), and among adolescents (11–15 years), according to the Health Behaviour in School-aged Children HBSC study, 16.6% of the examined were overweight, and 4.7% were obese [6,7]. The HBSC study revealed a significant increase in obesity among Polish adolescents in all age groups and in both genders, with higher prevalence rates in adolescents of the lowest socioeconomic background. Obese children are exposed to short and long-term health consequences, such as dyslipidemia, hypertension, insulin resistance or impaired glucose metabolism, increasing the risk of non-communicable diseases such as type 2 diabetes, cardiovascular disease, and specific types of cancer in adult life [8]. In Poland, both general and abdominal obesity in children has demonstrated a significant increase over last 46 years, with a tendency for a greater increase in the number of children with central obesity. Because visceral fat is more associated with a higher risk of metabolic complications, the number of Polish children and adolescents with type 2 diabetes mellitus, hypertension, non-alcoholic fatty liver disease and dyslipidemia is likely to increase in the future [9,10]. This situation is alarming and requires systemic changes in the approach to the prevention and treatment of obesity in children and adolescents.

Despite hereditary, environmental, cultural, and socioeconomic factors, which all play a role in the development of obesity, the most important factors are diet and physical activity. Diets rich in energy-dense food, too-large portion sizes and sedentary lifestyles lead to the accumulation of fat tissue due to a chronic imbalance between energy intake and energy expenditure [11]. That is why guidelines for the prevention and treatment of childhood obesity recommend interventions focused on every lifestyle factor in an age-appropriate way [11,12,13]. According to the recommendations of scientific societies, the goal of treatment of obesity in children is not only the loss of weight but also a permanent change in the family lifestyle, which is unfortunately difficult because the process is spread over time. The results of the studies show the beneficial role of interventions in overweight/obese children and adolescents, but the effect size depends on the methods applied. One systematic review of randomized trials estimating the efficacy of nonsurgical interventions in overweight/obese children showed a small-to-moderate effect of combined lifestyle interventions on body mass index (BMI); the largest effects were observed when lifestyle modifications were implemented with parental involvement [14]. Another systematic review of pediatric trials also stated that, in comparison to standard care or self-help, family-based lifestyle interventions aimed at changing dietary, behavioral and physical activity patterns can lead to a reduction in overweight [15]. The next systematic review found that interventions including diet, physical activity and behavioral change may be beneficial in achieving reductions of BMI and body mass in children from 6 to 11 years of age, and evidence suggests a very low incidence of adverse events [16]. Unfortunately, there is no standardized intervention scheme aimed at reducing body mass in children and adolescents in Poland. Therefore, the aim of this study was to evaluate whether an intensive intervention based on diet and physical activity has a better impact on the axiological and biochemical parameters than standard care.

## 2. Material and Methods

### 2.1. Study Group

The study group consisted of 20 children (6 boys, 14 girls), the range of age was 6.1–11.4 (mean age 8.9 ± SD 1.4) before puberty (Tanner stage 1) with obesity (mean body mass index standard deviation score-BMI SDS 3.76, SD 1.6). More than half of the study group lived in the countryside. All members of the study group demonstrated a similar range of initial physical activity, which was low or moderate, and none of these children attended additional sports activities after school. The participants were randomly assigned to two groups (Figure 1). The first group (I) started treatment with intensive intervention, and the second group (II) started with standard intervention. After 3 months of treatment, the groups swapped places. Exclusion criteria were: Overt metabolic complications of obesity (hypertension or diabetes mellitus) except abnormal lipid values and fatty liver features in an ultrasound examination, diagnosed abnormalities of the endocrine system, or a lack of consent to participate in the study. The criteria for the IDEFICS study were used to assess dyslipidemia in the study group [17]. According to these criteria, in the intensive group, 4 children were found to have abnormal lipid values (2 boys, 2 girls). In the case of one child, it was an elevated concentration of LDL-cholesterol and total cholesterol. In the case of the other three children, it was an increased concentration of triglycerides. As far as the standard group is concerned, 3 children (girls) had abnormal lipid values. The first child had an elevated level of LDL cholesterol and total cholesterol, the second had increased triglyceride concentration, and the third had reduced HDL concentration. None of the study participants met the criteria of the metabolic syndrome [18].

### 2.2. Dietetic Intervention in the Intensive Intervention Phase

At the first visit (60 min), the dietician (A.K-K) assessed the current diet based on the 24-h nutritional interview. On this basis, the correction of the diet was made, mainly concerning: Limiting the amount of simple sugars in the diet and high-fat products, fixing regular meal times, taking into account the lifestyle of families with special emphasis on breakfast, paying attention to the proportions of individual nutrients in the meal, and the recommended amount of water. The established diets had a calorific value appropriate to the child’s age in the range of 1200–1500 kcal, consisting of protein (5%–15%), carbohydrates (5%–65%), fat (30%; <10% saturated fatty acids, polyunsaturated up to 10%, monounsaturated up to 15%), and fiber (range: Age (year) plus 5 g – age (year) plus 10 g) [19]. At subsequent control visits, which took place every 2–3 weeks, diets were discussed on the basis of dietary records, modifications of the diet were introduced, and the progress of each child and family was monitored. Though the program was not based on any specific psychological strategies, the dietician and doctors continuously tried to give the children a positive motivation for lifestyle change.

### 2.3. Dietetic Intervention in the Standard Intervention Phase

Patients received a diet appropriate to their age and recommendations for modifying dietary behaviors the same as the patients in the intensive group, but there were no follow-up visits nor dietary modifications made on a current basis.

### 2.4. Physical Activity Recommendations in the Intensive Intervention Phase

At every visit, all participants underwent two motor and fitness tests. These included the standing broad jump (explosive muscular strength) and crunches for 30 s (abdominal muscular endurance). Data were obtained using the procedures described in the Eurofit Test Handbook, adjusted for obese children [20]. All children received encouragement from the investigators in order to achieve maximum performance. Everyone received the recommendation to do exercises every day at home. During the first week: 10 crunches, 10 sit-ups and 5 min of aerobic exercise—running or cycling. Each week during the intervention, the following were added: 1 additional crunch, 1 sit-up, and 1 min of aerobic exercise. Participants and their families were asked to keep a diary and a written confirmation of their exercises.

### 2.5. Physical Activity Recommendations in the Standard Intervention Phase

All children and parents received encouragement from the investigators in order to increase daily physical activity, and 60 min of exercise per day were recommended. No diary or other forms of control were used.

### 2.6. Anthropometric Measurements

Body weight and height were measured to the nearest 0.1 kg and 0.1 cm, respectively, using a stadiometer and a balanced scale. As the standard of reference, the normal values of the local population were used. In order to assess nutritional status, BMI interpretations of OLAF project (growth references for Polish school-aged children and adolescents) percentile charts were used [21]. Body composition was assessed by the bioelectrical impedance analysis (BIA) using a multifrequency bioelectrical impedance analyzer (Tanita BC 418 S MA, Tokyo, Japan). The measurements were performed according to the manufacturer’s guidelines at least 2 h after the ingestion of a light breakfast and urination. The following data were collected—fat mass (kg), fat percent in the whole body (fat %), fat-free mass (FFM) (kg), and total body water (TBW) (kg) [22]. The anthropometric measurement and BIA were assessed at each visit to the dietician.

### 2.7. Biochemical Assays, Ultrasound and Blood Pressure Examinations

Fasting insulin concentrations were measured using an ADVIA Centaur^®^ XP. Another biochemical analysis was performed in the fasting blood sample by the dry chemistry method with the Vitros 5.1.FF machine (Ortho-Clinical Diagnostics, Rochester, NY, USA). The estimated glomerular filtration rate (eGFR) was calculated by an on-line calculator based on the Schwartz and Counahan–Barratt methods adjusted for the pediatric population [23]. HOMA-IR was calculated using the formula: [fasting insulin level (µIU/mL) × fasting glucose level (mmol/L)]/22.5. The assessment was done at the beginning of the study and after the completion of each phase of the study.

A casual blood pressure measurement was done by the auscultatory method using a standard clinical sphygmomanometer with a cuff appropriate to the size of the child’s upper right arm. It was repeated three times during three different ambulatory visits. Systolic blood pressure SBP and diastolic blood pressure DBP were calculated as the mean value of these three results [24].

An ultrasound assessment of the liver was performed in all patients using a Voluson 730, (GE Medical System) with 8.5 and 3.5 MHz probes. One US-certified operator (DJ) performed the tests. The examiner was blinded to the clinical and laboratory data of patients.

### 2.8. Statistics

Categorical variables or categorized continuous variables were expressed as counts and percentages. In order to compare the two groups, the two-sided Mann–Whitney U-test and ANOVA tests were applied. To compare matched samples, the Wilcoxon signed-rank test was used. A *p*-value less than 0.05 was considered an indication of a statistically significant result. All statistical analyses were performed using the Statistica 12 PL software. (StatSoft Cracow, Poland).

### 2.9. Ethics

All parents of participants gave their informed consent for inclusion before they participated in the study. The study was conducted in accordance with the Declaration of Helsinki, and the protocol was approved by the Jagiellonian University Medical College Committee of Bioethics—Decision number 1072.6120.231.2017.

## 3. Results

The comparison of two interventions indicates the superiority of the effectiveness of intensive intervention in the improvement of anthropometric parameters. The decrease of the mean % of weight-to-height excess was significantly higher during the intensive phase of intervention (−9.8 vs. −1.6, *p* = 0.02) (Table 1).

Interestingly, the effect was more spectacular in the group that began with the intensive intervention phase (Table 2). Differences between the impact of two types of interventions on the analyzed biochemical parameters were less clear. The improvement of most of the analyzed parameters (except for insulin concentration, HOMA IR index and LDL cholesterol) was obtained after intensive intervention, although the differences were not statistically significant (Table 2).

The detailed analysis of the results of each form of intervention is presented in Table 2. The significant reduction of the mean % of weight-to-height excess, BMI SDS, body fat (BF%), and the time required to perform 30 crunches was observed in both groups after the first phase, regardless of the type of intervention. An additional significant improvement in the mean BMI SDS, body fat (BF%),and hip circumference occurred after the second phase only in the group in which an intensive intervention was used at this stage. In both groups, there was a significant positive effect of the intensive intervention on the HDL cholesterol level. Additionally, in Group I, there was a significant decrease in alanine transaminase ALT activity after intensive intervention.

## 4. Discussion

Standard intervention, such as recommendations for modifying dietary behaviors, is more common in daily clinical practice than a treatment based on lifestyle intervention involving specialist team-work (pediatrician and dietician). The biggest barrier in this approach to obesity treatment is that the dietetic consultation in the Polish public health care system is non-refundable. Notwithstanding, many guidelines for the management of child obesity indicate an important role of dieticians in this treatment [13,25]. Some studies have shown that interventions for where patients receive both, nutritional and lifestyle recommendations from well qualified dieticians are more effective than only standard recommendations given by their physician [26,27]. Moreover, obesity treatment should be targeted on a lifestyle changing, not only a diet modification, because physical activity and ways of spending free time are also very important and should be highlighted during treatment. The age of children when the lifestyle intervention should be taken is also important. It has been well established that in the crucial, prepubertal period, adiposity increases after its nadir in childhood (the so-called ‘adiposity rebound’). Children with a higher childhood BMI have a higher BMI, waist, and hip circumference in adulthood. A higher BMI in childhood may result in an earlier puberty and a subsequently earlier cessation of growth, with a consequent interruption to lean body mass acquisition [28]. One systematic review with a meta-analysis reported a positive effect of the lifestyle intervention as compared with usual care or minimal interventions (educational materials only) [29]. The overall effect size in the meta-analysis, which included seven studies (586 participants), was a decrease in the BMI of 1.30 kg/m^2^ at the end of an intensive intervention (95% CI: 1.03–1.58, heterogeneity I^2^ = 0%–48%). Four studies reported a BMI z-score change after the intensive intervention, with the pooled weight loss being a 0.09 BMI Z-score greater (0.02 to 0.15, I^2^ < 40%) in the lifestyle intervention compared with usual care. Similar observations were obtained for percentage body fat change, with the lifestyle intervention group losing 3.2% more body fat (95% CI: 1.39–5.01) than the usual care group. Studies with longer intervention periods (>6 months) showed a greater weight loss than shorter term interventions. In this review, four studies followed up participants at seven months to one year from the baseline, and the pooled results indicate that weight loss was sustained after program completion [29]. In younger children (two-to-five years old), similar results were obtained. In one study, there was an assessed maintenance of improved weight outcomes in preschoolers with obesity after 6 and 12 months following a randomized clinical trial comparing a home- and clinic-based behavioral intervention (Learning about Activity and Understanding Nutrition for Child Health [LAUNCH]) to motivational interviewing and standard care. This study confirmed the effectiveness of multicomponent behavioral treatment in reducing obesity in preschoolers and the chance of this intervention in maintaining improvements in weight after treatment ends. The children in LAUNCH showed a significantly greater decrease in average daily caloric intake at posttreatment and the 6 and 12 month follow-up compared with motivational interviewing and standard care, and they lost significantly more weight (−2.7 kg) compared with children in motivational interviewing (+1.3 kg) and standard care (+0.60 kg) from baseline to posttreatment [30].

In our study, regardless of the type of intervention, an improvement of anthropometric parameters was observed, but when we compare two interventions, we can see the superiority of the effectiveness of the intensive intervention. The mean changes in anthropometric parameters were higher after intensive intervention, and the mean weight-to-height excess and hip circumference decreased significantly. In the group starting with intensive intervention, the biggest decreases of body fat (BF%), (Δ = −3.6) and mean waist circumference (Δ = −3.8) were observed. Body fat (especially visceral fat) is strongly associated with atherogenic dyslipidemia seen in metabolic syndrome and insulin resistance, and its reduction is the most wanted change in obesity therapy [31].

The study results show positive effects of obesity treatment on metabolic parameters. Lifestyle interventions including diet modification and physical activity reported decreased insulin and/or HOMA-IR after one year of intervention [32,33]. Such results were not observed in the present study. This may be due to several reasons, including the low repeatability of insulin results in this age group [34]. Therefore, it seems more justified to assess the secondary biochemical parameters of insulin resistance than the insulin concentration level itself. The diet-only intervention caused greater reductions in the levels of triglycerides (at the end of active intervention) and low-density lipoprotein cholesterol (at subsequent follow-up) [35]. Study results suggested that diet plus exercise interventions may produce a greater improvement in metabolic parameters, especially HDL, cholesterol than only diet interventions [33,36]. In our study, positive changes were observed in all biochemical parameters regardless of the type of intervention, which is probably an overall effect of changes in body composition. However, in a comparison of group depending on the type of intervention, there was a positive significant change in the HDL cholesterol level. Dyslipidemia in obesity has been shown to be very responsive to even small changes in weight status, diet composition, and activity. Most importantly, in obese children and adolescents, even a small weight reduction is associated with significant decreases in triglycerides TG levels and increases in HDL–C levels [36,37]. A meta-analysis of five studies, including 440 participants between 8 and 16 years old, showed that lifestyle intervention had a significantly greater impact on the total cholesterol improvement compared with usual care both in the short-term (weighted mean difference WMD −0.40 mmol/L, 95% CI: −0.51 to −0.30; the heterogenity I^2^ = 0%; study length: four-to-six months) [15].

The rise of physical activity as an element of the lifestyle change is crucial in obesity reduction interventions. Physical activity does not just improve body composition and metabolic parameters, it also improves overall body fitness, which affects the quality of life of the patient. Pastucha D et al. compared the differences in anthropometric parameters, maximal oxygen uptake (VO2max) and physical activity (PA) between groups of 146 obese boys and 128 obese girls. The analyses showed significant increases in VO2max values in both groups of boys (*p* = 0.00028) and girls (*p* = 0.00012) [35]. Both groups also showed significant increases in the total amount of PA. In our study, the VO2max was not measured, but we assessed changes in explosive muscular strength and abdominal muscular endurance using two motor and fitness tests. Our observations were very similar, as we observed improvement of physical fitness (jump length, time required for 30 crunches) in children, regardless of the type of intervention.

While analyzing the effects of these two interventions, we should emphasize the advantage of intensive intervention that takes place from the first visit. We observed that the children who started with intensive intervention achieved better weight reduction results in comparison to children who had a standard intervention first. This may result from an extension of nutritional recommendations during intensive meetings with the dietician and a better motivation at the beginning of the intervention, which is maintained during the controls visits. This confirms that intensive care may be more effective than a one-off standard visit.

Limitations of the study: Our sample size was small, and, thus, the statistical power of the study was limited. This requires further research on larger groups.

## 5. Conclusions

In conclusion, the prevention and effective treatment of pediatric obesity should be goals of public health. This study confirmed that a complex intervention, including personalized dietitian counselling and physician care, is the most effective way for children to lose weight. Nevertheless, this study also showed that the effectiveness of children’s weight reduction may depend on the intensity of the intervention. To achieve good results, one piece dietary advice is not, enough because definitely better results were achieved through the intensive therapy based on systematic control visits than a one-off standard visit.

## Figures and Tables

**Figure 1 ijerph-16-02841-f001:**
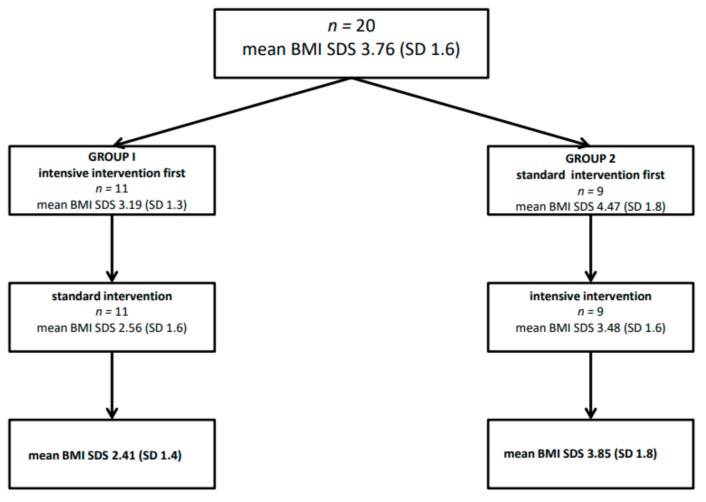
Reduction of body mass index BMI (SDS) in both arms of the study.

**Table 1 ijerph-16-02841-t001:** Comparison of the results of intensive intervention and standard intervention. Δ—change of the parameter value during the intervention.

Mean Change Value [SD]	Intensive Intervention	Standard Intervention	*p*
Δ% of weight-to-height excess	−9.8 [11.3]	−2.6 [6.5]	0.02 *
ΔBMI SDS	−0.5 [0.5]	−0.2 [0.6]	0.09
Δ body fat (BF%)	−2.9 [2.4]	−1.4 [2.4]	0.09
Δ waist circumference (cm)	−2.2 [5.6]	−1.3 [5.0]	0.9
Δ hip circumference (cm)	−2.4 [2.9]	0 [2.9]	0.01 *
Δ jump length (cm)	1 [17]	12 [17]	0.03 *
Δ time required for performing 30 crunches (s)	−4.3 [6.1]	−3.3 [6.7]	0.9
Δ systolic blood pressure (mmHg)	−3.8 [11.1]	−0.6 [9.5]	0.3
Δ diastolic blood pressure (mmHg)	−1.8 [5.0]	1.1 [5.0]	0.2
Δ fasting blood glucose (mmol/L)	−0.02 [0.3]	0.08 [0.5]	0.2
Δ fasting blood insulin (µIU/mL)	1.3 [8.6]	1.0 [12.0]	0.5
Δ HOMA IR	0.36 [1.8]	0.05 [3.5]	0.7
Δ fasting ALT activity (IU/L)	−7.2 [14.6]	−4.4 [15.2]	0.7
Δ fasting AST activity (IU/L)	−2.9 [9.7]	−1.4 [9.5]	0.5
Δ fasting GGT activity (IU/L)	−1.1 [2.6]	0.6 [5.2]	0.4
Δ fasting creatinine level (µmol/L)	−0.26 [2.5]	0.3 [3.6]	0.7
Δ eGFR	2.3 [6.2]	1.0 [9.2]	0.8
Δ fasting uric acid (µmol/L)	−3.1 [48.8]	2.2 [40.5]	0.8
Δ fasting total cholesterol (mmol/L)	−0.1 [0.5]	0.08 [0.6]	0.5
Δ fasting LDL (mmol/L)	−0.1 [0.4]	−0.1 [0.4]	0.9
Δ fasting HDL (mmol/L)	0.07 [0.1]	0.03 [0.1]	0.2
Δ fasting TG (mmol/L)	−0.2 [0.6]	0.1 [0.4]	0.07

**Table 2 ijerph-16-02841-t002:** Comparison of the results of each phase of intervention (intensive and standard) in both groups.

	Group I				Group II			
	Baseline	After Intensive Intervention (1st Phase)	After Standard Intervention (2nd Phase)	*p* Value	Baseline	After Standard Intervention (1st Phase)	After Intensive Intervention (2nd Phase)	*p* Value
mean % of weight-to-height excess [SD]	46.3 [18.7]	33 [22.7] (Δ = −13.3)	29.4 [19.4] (Δ = −3.6)	p = 0.003 * p = 0.9 **	58.8 [13.3]	51 [19] (Δ = −7.9)	45.6 [16.2] (Δ = − 5.4)	p = 0.03 * p = 0.08 **
mean BMI SDS	3.1 [1.3]	2.6 [1.6] (Δ = −0.5)	2.4 [1.4] (Δ = −0.2)	p = 0.006 * p = 0.4 **	4.5 [1.8]	3.8 [1.6] (Δ = −0.8)	3.5 [1.6] (Δ = −0.3)	p = 0.02 * p = 0.007 **
mean body fat (BF%)%	35 [6.4]	31.4 [5.9] (Δ = −3.6)	30 [5.7] (Δ = −1.4)	p = 0.005 * p = 0.5 **	38.6 [3.9]	36.2 [4.7] (Δ = −2,4)	34 [5.2] (Δ = −2,2)	p = 0.02 * p = 0.007 *
mean waist circumference (cm)	81 [12]	77.2 [10.7] (Δ = −3.8)	76 [10.4] (Δ = 1.2)	p = 0.1 * p = 0.5 **	84 [7.2]	80.5 [0.4] (Δ = −3.5)	81 [9.4] (Δ = +0,5)	p = 0.08 * p = 0.07 **
mean hip circumference (cm)	87.9 [11.4]	85.2 [12] (Δ = −2.7)	85.3 [11.1] (Δ = +0.1)	p = 0.04 * p = 0.06	89.2 [6.7]	88 [8.4] (Δ = −1.2)	85.9 [8.6] (Δ = −2.1)	p = 0.3 * p = 0.01 **
mean jump length (cm)	116.4 [18.2]	113.4 [19.8] (Δ = −3)	127.1 [15] (Δ = +13.7)	p = 0.2 * p = 0.02 **	101.2 [14.7]	117 [24.8] (Δ = +16.2)	124 [12.6] (Δ = +7)	p = 0.05 * p = 0.3 **
mean time required for performing 30 crunches (s)	37 [16.2]	34 [12.5] (Δ = −3)	31 [7.9] (Δ = −3)	p = 0.03 * p = 0.4 **	40 [4.1]	35.2 [6.7] (Δ = −4.8)	31 [3.2] (Δ = −4.2)	p = 0.04 * p = 0.06 **
systolic blood pressure (mmHg)	105.1 [10.2]	101.8 [8.7] (Δ = −3.3)	101.1 [3.3] (Δ = −0.7)	p = 0.3 * p = 0.9 **	106.6 [11.4]	105.5 [8.8] (Δ = −1.1)	102.2 [8.3] (Δ = −3.3)	p = 0.8 * p = 0.3 **
diastolic blood pressure (mmHg)	63.2 [4.6]	60 [0.1] (Δ = −3.2)	61.1 [3.3] (Δ = 1.1)	p = 0.06 * p = 0.9 **	62.2 [5.1]	63.3 [5] (Δ = +1.1)	63.3 [5] (Δ = 0)	p = 0.6 * p = 0.9 **
fasting blood glucose (mmol/L)	4.5 [0.4]	4.4 [0.2] (Δ = −0.1)	4.7 [0.3] (Δ = +0.3)	p = 0.8 * p = 0.1 **	4.6 [0.7]	4.6 [0.3] (Δ = 0)	4.6 [0.3] (Δ = 0)	p = 0.4 * p = 0.8 **
fasting blood insulin (µIU/mL)	13.5 [5.6]	12.7 [7.1] (Δ = −0.8)	16.9 [12.5] (Δ = −0.8)	p = 0.4 * p = 0.1 **	21.5 [14.6]	13.6 [7.1] (Δ = −7.9)	18.7 [15.7] (Δ = +5.1)	p = 0.3 * p = 0.6 **
HOMA IR	2.6 [0.9]	3.6 [3.2] (Δ = +1)	1.6 [2.9] (Δ = −2)	p = 0.6 * p = 0.2 **	4.6 [3.9]	2.8 [1.6] (Δ = −1.8)	3.9 [3.4] (Δ = +1.1)	p = 0.03 * p = 0.1 **
fasting ALT activity (IU/L)	32.3 [6.2]	25.7 [5.9] (Δ = −6.6)	25 [7.8] (Δ = −0.7)	p = 0.01 * p = 0.8 **	45.4 [10.9]	38.6 [25.7] (Δ = −6.8)	30.7 [7.7] (Δ = −7.9)	p = 0.1 * p = 0.4 **
fasting AST activity (IU/L)	29.1 [4.5]	27.8 [2.8] (Δ = −1.3)	27.7 [4.0] (Δ = −0.1)	p = 0.6 * p = 0.3 **	33.3 [8.2]	33.3 [16.6] (Δ = 0)	28.5 [4.3] (Δ = −4.8)	p = 0.2 * p = 0.4 **
fasting GGT activity (IU/L)	18 [6.4]	16 [5.1] (Δ = −2)	17.4 [10.6] (Δ = +1.4)	p = 0.2 * p = 0.32 **	18.8 [3.8]	17.2 [3.8] (Δ = −1.6)	16.9 [4.5] (Δ = −0.3)	p = 0.08 * p = 0.1 **
fasting creatinine level (µmol/L)	43.6 [4.8]	42.9 [5.0] (Δ = −0.7)	45.2 [6.4] (Δ = +2.3)	p = 0.4 * p = 0.8 **	41.7 [3.9]	41.5 [3.4] (Δ = −0.2)	42.4 [4.0] (Δ = +0.9)	p = 0.5 * p = 0.9 **
eGFR	118.5 [16.8]	122.3 [19.7] (Δ = +3.8)	117.8 [16.5] (Δ = −4.5)	p = 0.08 * p = 0.6 **	121.6 [12.5]	123 [8.2] (Δ = +1.4)	124 [9.9] (Δ = +1)	p = 0.6 * p = 0.9 **
fasting uric acid (µmol/L)	286.2 [76.2]	266.3 [76] (Δ = −19.9)	270.9 [89.1] (Δ = +4.6)	p = 0.5 * p = 0.8 **	322.5 [37.1]	319.9 [51.6] (Δ = −2.6)	325.2 [31.4] (Δ = +5.3)	p = 0.9 * p = 0.4 *
fasting total cholesterol (mmol/L)	4.5 [0.9]	4.3 [0.6] (Δ = −0.2)	4.5 [0.6] (Δ = +0.2)	p = 0.1 * p = 0.3 **	4.4 [0.6]	4.3 [0.6] (Δ = −0.1)	4.4 [0.6] (Δ = +0.1)	p = 0.6 * p = 0.5 **
fasting LDL (mmol/L)	2.8 [0.7]	2.6 [0.6] (Δ = −0.2)	2.5 [0.5] (Δ = −0.1)	p = 0.1 * p = 0.7 **	2.8 [0.5]	2.5 [0.6] (Δ = −0.3)	2.5 [0.5] (Δ = 0)	p = 0.7 * p = 0.2 **
fasting HDL (mmol/L)	1.1 [0.2]	1.2 [0.2] (Δ = +0.1)	1.3 [0.3] (Δ = +0.1)	p = 0.04 * p = 0.2 **	1.1 [0.3]	1.1 [0.4] (Δ = 0)	1.2 [0.3] (Δ = +0.1)	p = 0.8 * p = 0.04 **
fasting TG (mmol/L)	1.4 [1.3]	1.1 [0.9] (Δ = −0.3)	1.4 [1.3] (Δ = +0.1)	p = 0.06 * p = 0.07 **	1.5 [0.5]	1.4 [0.6] (Δ = −1)	1.3 [0.5] (Δ = −1)	p = 0.5 * p = 0.7 **

*p* value * after 1st phase, ** after 2nd phase (Δ = change value).

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
