# Peer review of "A Comparison of the Impact of Two Methods of Nutrition-Behavioral Intervention on Selected Auxological and Biochemical Parameters in Obese Prepubertal Children—Crossover Preliminary Study"

_ijerph, 2019, doi:10.3390/ijerph16162841_

Round 1

Reviewer 1 Report

I believe the authors have made some great strides in trying to fix the English in the paper. I think the introduction could be a little more robust with all the current theories on childhood obesity and why they have focused on diet and exercise with more relevant references.  

Author Response

We added to the introduction in lines 34-42 the epidemiological contex to better understanding the polish situation. And more information about risk factors and consequences of obesity in lines 44-46 and 53-55. We also added some newer references.

The manuscript has been checked and improved on British English.

Reviewer 2 Report

Even the topic of the manucsript interesting, the manuscript has some limitation (e.g the study is small amount of study subjects, only 20 kids) etc.

Author Response

We are aware of the limitations of small group size and we wrote about this issue in manuscript section "limitation of the study". The title also indicates a pilot study.

Reviewer 3 Report

Comments to Authors 

            The current study has showed that complex intervention based on systematic control visits, including personalized dietitian counseling and physician care, during the weight reduction process is more effective than a one-off standard visit.

           Authors are kindly requested to emphasize the current concepts about these issues in the context of recent knowledge and the available literature. This articles should be quoted in the References list.

References

Design of an educational strategy based on Intervention Mapping for nutritional health promotion in Child Care Centers. Eval Program Plann. 2019 Jun 12; 76: 101672. doi:10.1016/j.evalprogplan.2019.101672. Maintenance Following a Randomized Trial of a Clinic and Home-based Behavioral Intervention of Obesity in Preschoolers. J Pediatr. 2019 Jun 20. pii: S0022-3476(19)30556-6. doi:10.1016/j.jpeds.2019.05.004.

Author Response

Thank you for indicating the publications we have not included in the text. After reading them, we added both, one in the introduction in the line no. 58. And the second one in the discussion in lines 216-226.
